# Cross-sectional study on public health knowledge among first-year university students in Japan: Implications for educators and educational institutions

**Miwa Sekine**[1,2]*, **David Aune**[1], **Shuko Nojiri**[2], **Makino Watanabe**[1,3], **Yuko Nakanishi**[4], **Shinobu Sakurai**[5], **Tomomi Iwashimizu**[6], **Yasuaki Sakano**[7], **Tetsuya Takahashi**[8], **Yuji Nishizaki**[1,2]

1 Division of Medical Education, Juntendo University Faculty of Medicine, Tokyo, Japan, 2 Medical Technology Innovation Center, Juntendo University Faculty of Medicine, Tokyo, Japan, 3 Department of Physiology, Juntendo University Faculty of Medicine, Tokyo, Japan, 4 Juntendo University School of Health and Sport Science, Chiba, Japan, 5 Department of Public Health Nursing, Juntendo University, Faculty of Healthcare and Nursing, Chiba, Japan, 6 Juntendo University School of Health Sciences and Nursing, Shizuoka, Japan, 7 Department of Radiological Technology, Juntendo University Faculty of Health Science, Tokyo, Japan, 8 Department of Physical Therapy, Juntendo University Faculty of Health Science, Tokyo, Japan

* miwa@juntendo.ac.jp

**Data Availability Statement:** All relevant data are within the paper and its Supporting Information files.

## Abstract

In recent years, there have been increasing knowledge gaps and biases in public health information. This has become especially evident during the COVID-19 pandemic and has contributed to the spread of misinformation. With constant exposure to disinformation and misinformation through television, the internet, and social media, even university students studying healthcare-related subjects lack accurate public health knowledge. This study aimed to assess university students' knowledge levels of basic public health topics before they started their specialized education. Participants in this cross-sectional study were first-year students from medical schools, health-related colleges, and liberal arts colleges. A self-administered electronic survey was conducted from April to May 2021 at a private university in Japan, comprising six colleges with seven programs. Data analysis, conducted from June to December 2022, included students' self-reported public health knowledge, sources of information, and self-assessment of knowledge levels. Among the 1,562 students who received the questionnaire, 549 (192 male [35%], 353 female [64.3%], and 4 undisclosed [0.7%]) responded to one question (participants' response rate for each question; 59.6%–100%). The results showed that students had limited public health knowledge, especially in sexual health topics, and 10% of students reported not learning in class before university admission the following 11 topics: two on Alcohol, Tobacco, and Other Drugs; eight on Growth, Development, and Sexual Health; and one on Personal and Community Health. These results indicate significant knowledge gaps and biases, as well as gender gaps, in public health education, especially in the area of sexual health, which may help educators and educational institutions to better understand and prepare for further specialized

**Funding:** Initials of the authors who received each award: MS Grant number: 18K13259 The full name of funder: JSPS KAKENHI URL of funder website: https://www.jsps.go.jp/english/e-grants/ The funders had no role in study design, data collection and analysis, decision to publish, or preparation of the manuscript.

**Competing interests:** The authors have declared that no competing interests exist.

education. The findings also suggest a need to supplement and reinforce the foundation of public health knowledge for healthcare majors at the time of university admission.

## Introduction

Public health knowledge is vital for a healthy society. In recent years, there has been a surge in the prevalence of sexually transmitted diseases (STDs) in industrialized countries [1–5], including Japan [6, 7]. This increase has been further exacerbated during the COVID-19 pandemic. The increase in STDs among females in their 20s and 30s, especially the prevalence of Chlamydia infections among females in their teens, is particularly notable in Japan [8, 9]. STD prevention is an integral part of public health, and it is taught in health and physical education in Japan. Historically, sexual education as an essential ingredient of health and physical education changed from "non-existent" to "teaching chastity" in 1949, after which sex-segregated sexual education was mainly concerned with physiological differences or merely teaching about menstruation, which further evolved to somewhat sex-integrated education in Japan only in recent years [10, 11]. Although strides have been made in addressing crucial subjects such as gender equality, social justice, reproductive rights, and public health education, female students still show preference for single-sex sexual education and female educators. In the presence of other sexes in the same room, sexual health can be uncomfortable to discuss or study [12], which makes conversations among them more challenging.

According to previous studies, there is a notable difference between the genders in their preferred sources of knowledge regarding sexual health. Female students' ideal source of information was reported to be school as the primary source, followed by parents, friends, and the Internet. In contrast, male students preferred school as the primary source, followed by the Internet, friends, and parents, suggesting that there might be a difference in sexual education needs at home between male and female students [13]. Although diverse information sources generally mean more accurate knowledge of sexual health [14], not all information sources provide accurate information [15, 16], especially information sources such as the internet. Internet-based information could potentially be harmful due to the presence of inaccurate, misleading, or incomplete information based on insufficient or unsubstantiated "scientific" evidence or even sensational, anecdotal views [16]. In fact, people with lower health literacy were more likely to use and trust health information from social media, blogs, or celebrity webpages [17].

Education regarding STD prevention and basic public health education is included in the standard curriculum guidelines issued by the Ministry of Education, Culture, Sports, Science, and Technology (MEXT) under Health and Physical Education in Japan [18, 19]. However, health and sexual health education are not necessarily required for enrollment in any college because health topics are not included in entrance exams, leaving the extent of public health topics to each educator's discretion prior to university enrollment. This possible lack of public health education also applies to students enrolled in healthcare-related universities.

Healthcare professionals are required to disseminate this information. The lack of or inaccurate knowledge of public health among healthcare professionals has a direct effect on the public. However, obtaining accurate health information is not necessarily easy for people of different social and economic backgrounds or even sexes. It has been reported that there are digital disparities, in which higher socio-economic status and higher education are factors positively associated with digital health information seeking, which could result from either a

disparity of access or a disparity of literacy and comprehension [15, 16, 20]. Gender gaps exist in education [21, 22]. Additionally, other factors affecting public health knowledge exist, including promoting abstinence-only education, the lack of sexual health education altogether, or even the discouraging of its provision [23, 24]. The disparities in public health knowledge have been accentuated and brought to the forefront amidst the backdrop of the COVID-19 pandemic. This global crisis has magnified the existing gaps and biases, leading to the proliferation of misinformation and exacerbating knowledge disparities. A prominent consequence of this phenomenon has been the emergence of vaccine hesitancy, despite the development and widespread distribution [25, 26] of COVID-19 vaccines [27]. With constant exposure to disinformation and misinformation through television, the Internet, and social media [15, 28], it is imperative that younger generations, especially future healthcare providers, are given accurate health information that is the best available at the time.

Since the establishment of health promotion in the 1986 Ottawa Charter by the World Health Organization in Ottawa, Canada [29], numerous studies, reports, and surveys have been conducted on sexual education worldwide. However, it is unknown whether future healthcare professionals will acquire adequate health and public health knowledge before enrolling in university.

In this study, we surveyed healthcare-related college students to assess their level of public health knowledge and evaluate whether they have adequate knowledge before proceeding to specialized education in healthcare-related universities. The findings of this study can be beneficial for educators to supplement the gap in students' essential knowledge, especially healthcare-related college students, in the aftermath of the COVID-19 pandemic.

## Materials and methods

### Study design and participants

We developed a self-administered questionnaire designed to assess and evaluate first-year university students' knowledge of public health topics and conducted a cross-sectional study at Juntendo University, Tokyo, Japan. In order to assess public health knowledge prior to university with minimal influence of health-related knowledge from university classes, we selected first-year students within two months of admission to Juntendo University. Juntendo University is a private university that focuses on health science, comprising six colleges with seven programs: medicine, health care and nursing, health science and nursing, health science/radiological technology, health science/physical therapy, health and sports science, and international liberal arts. Table 1 presents the program characteristics.

We distributed web-based questionnaires to the entire cohort of first-year students (n = 1,562) across seven programs at Juntendo University. We utilized the university's universal message system to distribute the questionnaires among the different program divisions: Medicine (n = 136), Healthcare and Nursing (n = 204), Health Science and Nursing (n = 127), Health Science/Physical Therapy (n = 121), Health Science/Radiological Technology (n = 121) Health and Sports Science (n = 608), and International Liberal Arts (n = 245). This survey was conducted between April and May 2021. All participants provided informed consent and had the option to decline participation. Prior to their involvement, participants received a comprehensive overview of the study, including detailed information about data management procedures. They were informed that their data would be anonymized and participation was voluntary. We obtained informed consent from respondents by asking to tick the checkbox in the survey and collected the completed questionnaires. Respondents who did not provide informed consent were excluded from the study. Parental consent was not required since first-year students in Japan are 18 years of age or older. There were no other exclusion criteria.

**Table 1. Basic characteristics of participants (n = 549).**

| Colleges | Characteristics of colleges | Total (%) | Sex | | |
|---|---|---|---|---|---|
| | | | Male (%) | Female (%) | Undisclosed (%) |
| **Medicine** | Private medical school established in 1838, headquarters in Tokyo | 74(13.5) | 37(50) | 37(50) | 0(0) |
| **Health Care and Nursing** | Nursing course serving greater Tokyo area | 63(11.5) | 1(1.6) | 61(96.8) | 1(1.6) |
| **Health Science and Nursing** | Nursing course serving in the local area community of Shizuoka prefecture | 90(16.4) | 6(6.7) | 84(93.3) | 0(0) |
| **Health Science/Physical Therapy** | Training course for physical therapists | 83(15.1) | 29(34.9) | 54(65.1) | 0(0) |
| **Health Science/ Radiological Technology** | Training course for radiological technologists | 30(5.5) | 9(30) | 20(66.7) | 1(3.3) |
| **Health and Sports Science** | Six courses related to sports. e.g., athlete, sports management, coaching, etc. | 159(29) | 94(59.1) | 64(40.3) | 1(0.6) |
| **International Liberal Arts** | Courses on global sociological issues, intercultural communication, global health service | 50(9.1) | 16(32) | 33(66) | 1(2) |
| **Total** | | 549(100) | 192(35) | 353(64.3) | 4(0.7) |

NOTE. Values in parenthesis represent percentage of valid total N.

## Questionnaire and survey

To examine the differences in students' awareness and knowledge of health and public health, we identified essential topics and public health knowledge from the Japanese curriculum guidelines, educational policies, educational contents of textbooks, and standard health education policies stipulated by the Centers for Disease Control and Prevention (CDC). We selected several crucial health education sections and topics shared by Japan and the United States: 1) Nutrition and Physical Activity; 2) Growth, Development, and Sexual Health; 3) Injury Prevention; 4) Alcohol, Tobacco, and Other Drugs; 5) Mental and Social Health; and 6) Personal and Community Health. We identified 31 knowledge topics for the educational content, which are listed in S1 Table. For each of these knowledge topics, we developed a set of three self-assessment questions, including: 1) the presence/absence of knowledge (acquired in class); 2) information sources for the knowledge; and 3) the level of knowledge on a 10-point scale ranging from 1 (no knowledge at all) to 10 (enough knowledge to teach someone else). In total, we prepared 86 self-assessment questions. We prepared five questions on knowledge of lifestyle-related diseases, contraception, LGBTQIA+ issues, pathogens causing infections, and transmission of infection, for a total of 91 questions (S2 Table). We developed the survey questions in consultation with medical and health education experts. We prepared a survey using Google Forms and administered it to all freshmen in all departments. All survey data was collected anonymously. The questionnaires included self-reported sex (i.e., male, female, undisclosed, or other). With a racially homogeneous environment and most university enrollment occurring immediately after high school, we did not ask about ethnicity or age to maintain anonymity. We consulted with two native English bilinguals and two native Japanese bilinguals to translate the questions and answers.

## Data analysis

Descriptive analyses were performed to examine the respondents' characteristics and responses. Frequencies and percentages were utilized for categorical variables, and means and standard deviations, or medians and 95% confidence intervals were utilized for continuous variables. If there was at least one answer to a question, the answer was accepted. Unless

otherwise stated, the percentage was calculated based on the number of responses to each question. The number of topics for which students reported having knowledge and self-assessment knowledge levels was calculated as a percentage of the total knowledge score and total knowledge self-assessment score, and normality was tested using the Shapiro-Wilk test. The maximum score for total knowledge and self-assessment are 29 and 290, respectively. We converted the total knowledge score and self-assessment score into percentage scores, with a maximum attainable score of 100%. To evaluate knowledge level and self-assessment levels across colleges, we categorized scores below the 25th percentile as "Poor," scores between the 25th and 75th percentile as "Fair," and scores at or above the 75th percentile as "Good." We examined the total knowledge and self-assessment scores, as well as the categorization of these scores into three levels ("Poor," "Fair," "Good"), using appropriate methods. Specifically, we employed Kruskal–Wallis tests for non-parametric variance analysis, performed one-way analysis of variance for parametric variance assessment, employed Pearson Chi-Square Tests for proportions, and conducted Bonferroni adjusted pairwise comparisons. These analyses aimed to evaluate whether statistically significant differences existed between colleges regarding knowledge and self-assessment levels. The undisclosed sex group was excluded from the examination based on sex comparison. The Pearson's chi-square test was used to examine sex-based differences. Z-tests were used to further explore the differences, and the Bonferroni correction was employed to adjust for multiple comparisons. Mann-Whitney U tests were used to examine the mean rank differences in self-assessment by sex. Given the available resources and constraints in data collection, we gathered data from 549 first-year students, representing a subset of the total population of 1,562 first-year students within the university. As such, a formal sample size calculation or power analysis was not conducted because of the exploratory nature of this study and the comprehensive distribution of the questionnaire to the total first-year student population at Juntendo University.

All statistical analyses were performed using IBM SPSS Statistics for Windows, Version 28.0 (Armonk, NY: IBM Corp.). The p-value threshold for significance was adjusted for multiple comparisons using the Bonferroni correction. The threshold for statistical significance was set at $p < .05$.

### Ethics statement

This study was approved by the Juntendo University Ethical Review Board (No. 2020324) and followed the Strengthening the Reporting of Observational Studies in Epidemiology Reporting Guidelines. We obtained informed consent from respondents and collected the completed questionnaires.

## Results

### Response

Among the 1,562 students who received the questionnaires, 549 (35.15%) responded to at least one question (response rate: total 35.1%, medicine 54.4%, healthcare and nursing 30.9%, health science and nursing 70.9%, health science/physical therapy 68.6%, health science/radiological technology 24.8%, health and sports science 26.2%, and international liberal arts 20.4%). The study included 192 male (35%) and 353 female students (64.3%). The participants' response rate ranged from a minimum of 59.6% to a maximum of 100% (S2 Table).

### Knowledge of public health topics total score and self-assessment total score

Kruskal–Wallis tests showed no significant differences among sexes [H(2) = 3.763, $p = .152$] or colleges in self-reported total knowledge scores among males [H(4) = 3.813, $p = .432$], females

**Table 2. Total score rate of (A) students' reported public health knowledge and (B) self-assessment.**

| | (A)Knowledge | | | | | | | | | (B)Self-assessment score | | | | | |
| --- | --- | --- | --- | --- | --- | --- | --- | --- | --- | --- | --- | --- | --- | --- | --- |
| | Male | | | Female | | | Total | | | Male | | Female | | Total | |
| | N | Median | 95% CI | N | Median | 95% CI | N | Median | 95% CI | N | Mean±SD | N | Mean±SD | N | Mean±SD |
| **Medicine** | 37 | 86.21 | 82.76–89.66 | 37 | 89.66 | 89.66–93.1 | 74 | 89.66 | 89.66–93.1 | 31 | 56.97 ±12.23 | 35 | 61.23 ±11.55 | 66 | 59.23 ±11.97 |
| **Nursing** | 7 | 89.66 | 86.21–100 | 145 | 86.21 | 86.21–89.66 | 153 | 86.21 | 86.21–89.66 | 7 | 61.53 ±11.13 | 132 | 56.01 ±13.39 | 140 | 56.16 ±13.34 |
| **Healthcare therapist** | 38 | 87.93 | 86.21–89.66 | 74 | 89.66 | 89.66–93.1 | 113 | 89.66 | 89.66–93.1 | 34 | 57.35±9.85 | 67 | 54.84 ±12.51 | 102 | 55.78 ±11.67 |
| **Health and sports** | 94 | 84.48 | 79.31–89.66 | 64 | 89.66 | 86.21–96.55 | 159 | 86.21 | 86.21–89.66 | 82 | 55.68 ±15.21 | 58 | 58.44±11.5 | 141 | 56.83 ±13.77 |
| **International liberal arts** | 16 | 79.31 | 79.31–93.1 | 33 | 89.66 | 86.21–96.55 | 50 | 86.21 | 79.31–89.66 | 14 | 57.17 ±16.85 | 33 | 55.09±14.4 | 48 | 55.93 ±14.93 |
| **Total** | 192 | 86.21 | 86.21–89.66 | 353 | 89.66 | 89.66–93.1 | 549 | 86.21 | 86.21–89.66 | 168 | 56.63 ±13.66 | 325 | 56.67 ±12.89 | 497 | 56.66 ±13.12 |

NOTE: The number of topics about which students reported having knowledge and their self-assessment knowledge level for the topic were calculated to percentage as score. N represents the number of students who answered all questions about self-assessment (S1 File).

[H(4) = 8.175, $p$ = .085], or total [H(4) = 2.186, $p$ = .702] (Table 2A). Additionally, according to the one-way analysis of variance results, there were no significant differences between sexes [F(2, 494) = 0.001, $p$ = .999] or colleges on the self-assessment total score among males [F(4, 163) = 0.353, $p$ = .842], females [F(4, 320) = 1.940, $p$ = .104], or total [F(4, 492) = 0.841, $p$ = .500] (Table 2B). Pearson Chi-Square Tests indicated no significant differences among colleges in the three levels of knowledge ($p$ = .122) or self-assessment ($p$ = .635). However, a significant difference was observed among sexes in the three levels of self-assessment in the Health and Sports College ($p$ = .011). Further pairwise comparison revealed significant differences in the "Poor" ($p$ = .010) and "Fair" ($p$ = .006) categories (Table 3).

## Knowledge of public health topics and classroom involvement

There were 28 questions about whether students had knowledge of the topic and if they had learned it in class. More than 10% of the students were unfamiliar with 12 topics (two for Alcohol, Tobacco, and Other Drugs; nine for Growth, Development, and Sexual Health; and one for Personal and Community Health) of the 28 topics (Table 4A). More than 10% of the students responded that they did not learn about 11 topics (two for Alcohol, Tobacco, and Other Drugs; eight for Growth, Development, and Sexual Health; and one for Personal and Community Health) in class (Table 4B).

## Knowledge of specific public health items

Questions on whether they had knowledge of specific items, diseases, or terms like lifestyle-related diseases, contraception, LGBTQIA+ issues, pathogens causing infections, and transmission of infection showed that more than 75% of students reported that they had knowledge of at least one or more specific items (S3 Table).

## Information sources for public health topics

We inquired about the sources of information for 29 topics and found that most students relied on teachers as their primary source of information. However, for 12 out of 29 topics, less than 75% of students chose teachers as their source of information. For nine out of 29 topics,

**Table 3. Three level of (A) students' reported public health knowledge and (B) self-assessment.**

| | | (A)Knowledge | | | (B)Self-assessment score | | |
|---|---|---|---|---|---|---|---|
| | | Male(%) | Female(%) | Total(%) | Male(%) | Female(%) | Total(%) |
| **Medicine** | Good | 7(9.5) | 11(14.9) | 18(24.3) | 8(12.1) | 12(18.2) | 20(30.3) |
| | Fair | 20(27) | 22(29.7) | 42(56.8) | 16(24.2) | 18(27.3) | 34(51.5) |
| | Poor | 10(13.5) | 4(5.4) | 14(18.9) | 7(10.6) | 5(7.6) | 12(18.2) |
| | Total | 37(50) | 37(50) | 74(100) | 31(47) | 35(53) | 66(100) |
| **Nursing** | Good | 2(1.3) | 22(14.4) | 24(15.7) | 3(2.1) | 32(22.9) | 35(25) |
| | Fair | 4(2.6) | 68(44.4) | 72(47.1) | 3(2.1) | 62(44.3) | 65(46.4) |
| | Poor | 1(0.7) | 55(35.9) | 57(37.3) | 1(0.7) | 38(27.1) | 40(28.6) |
| | Total | 7(4.6) | 145(94.8) | 153(100) | 7(5) | 132(94.3) | 140(100) |
| **Healthcare therapist** | Good | 5(4.4) | 16(14.2) | 21(18.6) | 6(5.9) | 13(12.7) | 19(18.6) |
| | Fair | 21(18.6) | 36(31.9) | 58(51.3) | 24(23.5) | 33(32.4) | 58(56.9) |
| | Poor | 12(10.6) | 22(19.5) | 34(30.1) | 4(3.9) | 21(20.6) | 25(24.5) |
| | Total | 38(33.6) | 74(65.5) | 113(100) | 34(33.3) | 67(65.7) | 102(100) |
| **Health and sports** | Good | 16(10.1) | 18(11.3) | 34(21.4) | 22(15.6) | 13(9.2) | 35(24.8) |
| | Fair | 40(25.2) | 30(18.9) | 71(44.7) | 33(23.4) | 37(26.2) | 71(50.4) |
| | Poor | 38(23.9) | 16(10.1) | 54(34) | 27(19.1) | 8(5.7) | 35(24.8) |
| | Total | 94(59.1) | 64(40.3) | 159(100) | 82(58.2) | 58(41.1) | 141(100) |
| **International liberal arts** | Good | 3(6) | 8(16) | 11(22) | 3(6.3) | 8(16.7) | 12(25) |
| | Fair | 4(8) | 13(26) | 18(36) | 7(14.6) | 15(31.3) | 22(45.8) |
| | Poor | 9(18) | 12(24) | 21(42) | 4(8.3) | 10(20.8) | 14(29.2) |
| | Total | 16(32) | 33(66) | 50(100) | 14(29.2) | 33(68.8) | 48(100) |
| **Total** | Good | 33(6) | 75(13.7) | 108(19.7) | 42(8.5) | 78(15.7) | 121(24.3) |
| | Fair | 89(16.2) | 169(30.8) | 261(47.5) | 83(16.7) | 165(33.2) | 250(50.3) |
| | Poor | 70(12.8) | 109(19.9) | 180(32.8) | 43(8.7) | 82(16.5) | 126(25.4) |
| | Total | 192(35) | 353(64.3) | 549(100) | 168(33.8) | 325(65.4) | 497(100) |

NOTE: Good: >75 percentile, Fair: 25–75 percentile, Poor: <25 percentile. The percentages represent the proportions relative to the total N of each college (S1 File).

more than 25% of the students relied on their parent(s) for information. While friends were not necessarily the primary source of information, more than 10% of students obtained information on six public health topics from friends. Additionally, the Internet was an important information source for all the topics. However, for six out of the 29 topics, more than 40% of the students selected the Internet as their primary source of information, with more than 50% selecting the Internet for two of the topics (Table 5).

## Self-assessment of knowledge level

The lowest level of self-assessed knowledge was 4.0 (95% CI, 4.0–5.0) for topics such as the health effects of e-cigarettes, women's basal body temperature, and LGBTQIA+ issues. Moreover, the pinnacle of self-assessed knowledge reached 7.0 (95% CI, 7.0–8.0) on a scale with a maximum score of 10.0 for topics such as the health effects of smoking, the health hazards of passive smoking, smoking dependence, the mental and physical effects of mind-altering drugs, and ovulation and menstruation.

## Comparison of previous knowledge and class involvement by sex

There was a significant difference between males and females in their previous knowledge of topics such as allergic reactions, alcohol consumption effects on pregnancy, ovulation and

**Table 4. Summary of students' previous knowledge regarding public health topics: (A) Did not know about the topic; (B) Did not learn about the topic in classes.**

| | (A) Did not know about the topic | | | | (B) Did not learn about it in classes | | | |
|---|---|---|---|---|---|---|---|---|
| | Total (N = 545) | Sex (%) | | *p* | Total (N = 545) | Sex (%) | | *P* |
| | | Male (N = 192) | Female (N = 353) | | | Male (N = 192) | Female (N = 353) | |
| *Nutrition and Physical Activity* | | | | | | | | |
| **Lifestyle-related diseases** | 46 (8.4) | 19 (9.9) | 27 (7.6) | .348 | 11 (2) | 7 (3.6) | 4 (1.1) | .044 |
| **Food poisoning** | 38 (7) | 15 (7.8) | 23 (6.5) | .570 | 21 (3.9) | 10 (5.2) | 11 (3.1) | .225 |
| **Allergic reaction** | 8 (1.5) | 6 (3.1) | 2 (0.6) | .018 | 7 (1.3) | 5 (2.6) | 2 (0.6) | .044 |
| *Alcohol, Tobacco, and Other Drugs* | | | | | | | | |
| **Health effect of smoking** | 1 (0.2) | 1 (0.5) | 0 (0) | NA | 0 (0) | 0 (0) | 0 (0) | NA |
| **Health hazard of passive smoking** | 10 (1.8) | 1 (0.5) | 9 (2.5) | .092 | 5 (0.9) | 1 (0.5) | 4 (1.1) | .475 |
| **Smoking effect on pregnancy** | 31 (5.7) | 15 (7.8) | 16 (4.5) | .114 | 21 (3.9) | 8 (4.2) | 13 (3.7) | .779 |
| **Dependence of smoking** | 6 (1.1) | 1 (0.5) | 5 (1.4) | .341 | 2 (0.4) | 0 (0) | 2 (0.6) | |
| **Health effect of e-cigarettes** | 197 (36.1) | 63 (32.8) | 134 (38) | .223 | 175 (32.1) | 58 (30.2) | 117 (33.1) | .470 |
| **Effect of alcohol consumption** | 9 (1.7) | 4 (2.1) | 5 (1.4) | .554 | 4 (0.7) | 2 (1) | 2 (0.6) | .531 |
| **Alcohol consumption effect on pregnancy** | 39 (7.2) | 20 (10.4) | 19 (5.4) | .030 | 24 (4.4) | 11 (5.7) | 13 (3.7) | .269 |
| **Mental/physical effect of mind-altering drugs** | 1 (0.2) | 1 (0.5) | 0 (0) | NA | 1 (0.2) | 1 (0.5) | 0 (0) | NA |
| **Alcohol, smoking and drug addiction** | 33 (6.1) | 12 (6.3) | 21 (5.9) | .888 | 17 (3.1) | 7 (3.6) | 10 (2.8) | .602 |
| **Addiction** | 117 (21.5) | 45 (23.4) | 72 (20.4) | .418 | 85 (15.6) | 31 (16.1) | 54 (15.3) | .805 |
| *Growth, Development and Sexual Health* | | | | | | | | |
| **Ovulation and menstruation** | 17 (3.1) | 15 (7.8) | 2 (0.6) | < .001 | 8 (1.5) | 7 (3.6) | 1 (0.3) | .002 |
| **Ejaculation** | 21 (3.9) | 1 (0.5) | 20 (5.7) | .003 | 10 (1.8) | 1 (0.5) | 9 (2.5) | .094 |
| **How to use condoms** | 129 (23.7) | 15 (7.8) | 114 (32.3) | < .001 | 86 (15.8) | 10 (5.2) | 76 (21.5) | < .001 |
| **Woman's basal body temperature** | 190 (34.9) | 103 (53.6) | 87 (24.6) | < .001 | 139 (25.5) | 76 (39.6) | 63 (17.8) | < .001 |
| **Measurement of woman's basal body temperature** | 274 (50.3) | 128 (66.7) | 146 (41.4) | < .001 | 217 (39.8) | 106 (55.2) | 111 (31.4) | < .001 |
| **Fertilization and pregnancy** | 7 (1.3) | 3 (1.6) | 4 (1.1) | .666 | 2 (0.4) | 1 (0.5) | 1 (0.3) | .659 |
| **Gestation period and childbirth** | 67 (12.3) | 30 (15.6) | 37 (10.5) | .085 | 24 (4.4) | 15 (7.8) | 9 (2.5) | .004 |
| **Public services related to pregnancy** | 96 (17.6) | 44 (22.9) | 52 (14.7) | .015 | 63 (11.6) | 32 (16.7) | 31 (8.8) | .006 |
| **Significance and importance of family planning** | 155 (28.4) | 60 (31.3) | 95 (26.9) | .293 | 118 (21.7) | 43 (22.4) | 75 (21.2) | .768 |
| **Abortion under Maternal Health Act in Japan** | 98 (18) | 46 (24) | 52 (14.7) | .008 | 67 (12.3) | 35 (18.2) | 32 (9.1) | .002 |
| **Sex and gender** | 131 (24) | 33 (17.2) | 98 (27.8) | .005 | 105 (19.3) | 27 (14.1) | 78 (22.1) | .022 |
| **LGBTQIA+** | 118 (21.7) | 41 (21.4) | 77 (21.8) | .862 | 96 (17.6) | 33 (17.2) | 63 (17.8) | .813 |
| *Personal and Community Health* | | | | | | | | |
| **Birth control pills and preventing sexually transmitted diseases** | 206 (37.8) | 73 (38) | 133 (37.7) | .940 | 163 (29.9) | 58 (30.2) | 105 (29.7) | .913 |
| **Prevention of sexually transmitted diseases** | 28 (5.1) | 12 (6.3) | 16 (4.5) | .383 | 16 (2.9) | 8 (4.2) | 8 (2.3) | .208 |
| **Vaccines** | 26 (4.8) | 11 (5.7) | 15 (4.2) | .428 | 21 (3.9) | 9 (4.7) | 12 (3.4) | .445 |

NOTE: Values in parenthesis represent percentage of valid N for each category. Results are based on two-sided tests for comparison between sexes. The p-value threshold for significance was adjusted for multiple comparisons using the Bonferroni correction. The threshold for statistical significance was set at p < .05 (S1 File).

menstruation, ejaculation, how to use condoms, women's basal body temperature, measurement of women's basal body temperature, public services related to pregnancy, abortion under the Maternal Health Act in Japan, and sex and gender (Table 4A). Moreover, there were significant differences between males and females in reporting that they did not learn about certain topics in their classes, such as lifestyle-related disease, allergic reactions, ovulation and

**Table 5. (A). Summary of sources of information regarding public health topics.** (B). Summary of sources of information regarding public health topics.

**A**

| | Teacher(s) as information source | | | | Parent(s) as information source | | | |
|---|---|---|---|---|---|---|---|---|
| | Total (N = 545) | Sex (%) | | p | Total (N = 545) | Sex (%) | | p |
| | | Male (N = 192) | Female (N = 353) | | | Male (N = 192) | Female (N = 353) | |
| **Nutrition and Physical Activity** | | | | | | | | |
| Lifestyle-related diseases | 467 (85.7) | 161 (83.9) | 306 (86.7) | .397 | 142 (26.1) | 58 (30.2) | 84 (23.8) | .100 |
| Food poisoning | 332 (60.9) | 114 (59.4) | 218 (61.8) | .596 | 230 (42.2) | 76 (39.6) | 154 (43.6) | .366 |
| Allergic reaction | 362 (66.4) | 121 (63) | 241 (68.3) | .357 | 262 (48.1) | 86 (44.8) | 176 (49.9) | .362 |
| **Alcohol, Tobacco, and Other Drugs** | | | | | | | | |
| Health effect of smoking | 491 (90.1) | 174 (90.6) | 317 (89.8) | .831 | 246 (45.1) | 83 (43.2) | 163 (46.2) | .491 |
| Health hazard of passive smoking | 481 (88.3) | 166 (86.5) | 315 (89.2) | .212 | 179 (32.8) | 65 (33.9) | 114 (32.3) | .761 |
| Smoking effect on pregnancy | 423 (77.6) | 152 (79.2) | 271 (76.8) | .658 | 112 (20.6) | 35 (18.2) | 77 (21.8) | .293 |
| Dependence of smoking | 470 (86.2) | 168 (87.5) | 302 (85.6) | .530 | 186 (34.1) | 72 (37.5) | 114 (32.3) | .222 |
| Health effect of e-cigarettes | 244 (44.8) | 88 (45.8) | 156 (44.2) | .685 | 103 (18.9) | 39 (20.3) | 64 (18.1) | .817 |
| Effect of alcohol consumption | 470 (86.2) | 168 (87.5) | 302 (85.6) | .637 | 207 (38) | 78 (40.6) | 129 (36.5) | .376 |
| Alcohol consumption effect on pregnancy | 435 (79.8) | 157 (81.8) | 278 (78.8) | .171 | 130 (23.9) | 43 (22.4) | 87 (24.6) | .636 |
| Mental/physical effect of mind-altering drugs | 501 (91.9) | 173 (90.1) | 328 (92.9) | .334 | 113 (20.7) | 44 (22.9) | 69 (19.5) | .338 |
| Alcohol, smoking and drug addiction | 480 (88.1) | 169 (88) | 311 (88.1) | .926 | 130 (23.9) | 61 (31.8) | 69 (19.5) | .001 |
| Addiction | 381 (69.9) | 136 (70.8) | 245 (69.4) | .705 | 75 (13.8) | 30 (15.6) | 45 (12.7) | .446 |
| **Growth, Development and Sexual Health** | | | | | | | | |
| Ovulation and menstruation | 475 (87.2) | 166 (86.5) | 309 (87.5) | .675 | 222 (40.7) | 23 (12) | 199 (56.4) | < .001 |
| Ejaculation | 453 (83.1) | 159 (82.8) | 294 (83.3) | .495 | 28 (5.1) | 12 (6.3) | 16 (4.5) | .417 |
| How to use condoms | 329 (60.4) | 130 (67.7) | 199 (56.4) | .854 | 9 (1.7) | 3 (1.6) | 6 (1.7) | .714 |
| Woman's basal body temperature | 273 (50.1) | 82 (42.7) | 191 (54.1) | .611 | 70 (12.8) | 7 (3.6) | 63 (17.8) | < .001 |
| Fertilization and pregnancy | 488 (89.5) | 172 (89.6) | 316 (89.5) | .794 | 44 (8.1) | 11 (5.7) | 33 (9.3) | .131 |
| Gestation period and childbirth | 460 (84.4) | 162 (84.4) | 298 (84.4) | .760 | 93 (17.1) | 18 (9.4) | 75 (21.2) | .001 |
| Public services related to pregnancy | 351 (64.4) | 121 (63) | 230 (65.2) | .843 | 193 (35.4) | 54 (28.1) | 139 (39.4) | .020 |
| Significance and importance of family planning | 331 (60.7) | 121 (63) | 210 (59.5) | .193 | 90 (16.5) | 31 (16.1) | 59 (16.7) | .931 |
| Contraception | 427 (78.3) | 154 (80.2) | 273 (77.3) | .548 | 35 (6.4) | 10 (5.2) | 25 (7.1) | .381 |
| Abortion under Maternal Health Act in Japan | 386 (70.8) | 135 (70.3) | 251 (71.1) | .751 | 28 (5.1) | 3 (1.6) | 25 (7.1) | .006 |
| Sex and gender | 343 (62.9) | 135 (70.3) | 208 (58.9) | .085 | 14 (2.6) | 5 (2.6) | 9 (2.5) | .918 |
| LGBTQIA+ | 299 (54.9) | 115 (59.9) | 184 (52.1) | .038 | 31 (5.7) | 12 (6.3) | 19 (5.4) | .645 |
| **Personal and Community Health** | | | | | | | | |
| Prevention of sexually transmitted diseases | 459 (84.2) | 167 (87) | 292 (82.7) | .081 | 31 (5.7) | 9 (4.7) | 22 (6.2) | .475 |
| Pathogens that cause infections | 430 (78.9) | 152 (79.2) | 278 (78.8) | .526 | 55 (10.1) | 18 (9.4) | 37 (10.5) | .745 |
| Transmission of infection | 409 (75) | 151 (78.6) | 258 (73.1) | .110 | 98 (18) | 30 (15.6) | 68 (19.3) | .307 |
| Vaccines | 438 (80.4) | 151 (78.6) | 287 (81.3) | .713 | 133 (24.4) | 45 (23.4) | 88 (24.9) | .785 |

**B**

| | Friend(s) as information source | | | | Internet as information source | | | |
|---|---|---|---|---|---|---|---|---|
| | Total (N = 545) | Sex (%) | | p | Total (N = 545) | Sex (%) | | p |
| | | Male (N = 192) | Female (N = 353) | | | Male (N = 192) | Female (N = 353) | |
| **Nutrition and Physical Activity** | | | | | | | | |
| Lifestyle-related diseases | 17 (3.1) | 12 (6.3) | 5 (1.4) | .002 | 208 (38.2) | 83 (43.2) | 125 (35.4) | .069 |

*(Continued)*

**Table 5.** (Continued)

| | | | | | | | | |
|---|---|---|---|---|---|---|---|---|
| Food poisoning | 25 (4.6) | 10 (5.2) | 15 (4.2) | .606 | 264 (48.4) | 109 (56.8) | 155 (43.9) | .004 |
| Allergic reaction | 61 (11.2) | 24 (12.5) | 37 (10.5) | .424 | 244 (44.8) | 99 (51.6) | 145 (41.1) | .010 |
| **Alcohol, Tobacco, and Other Drugs** | | | | | | | | |
| Health effect of smoking | 45 (8.3) | 24 (12.5) | 21 (5.9) | .008 | 233 (42.8) | 98 (51) | 135 (38.2) | .004 |
| Health hazard of passive smoking | 33 (6.1) | 26 (13.5) | 7 (2) | < .001 | 187 (34.3) | 80 (41.7) | 107 (30.3) | .009 |
| Smoking effect on pregnancy | 14 (2.6) | 9 (4.7) | 5 (1.4) | .022 | 152 (27.9) | 62 (32.3) | 90 (25.5) | .105 |
| Dependence of smoking | 45 (8.3) | 29 (15.1) | 16 (4.5) | < .001 | 173 (31.7) | 78 (40.6) | 95 (26.9) | .001 |
| Health effect of e-cigarettes | 39 (7.2) | 24 (12.5) | 15 (4.2) | .001 | 166 (30.5) | 74 (38.5) | 92 (26.1) | .009 |
| Effect of alcohol consumption | 49 (9) | 26 (13.5) | 23 (6.5) | .007 | 182 (33.4) | 75 (39.1) | 107 (30.3) | .043 |
| Alcohol consumption effect on pregnancy | 19 (3.5) | 10 (5.2) | 9 (2.5) | .096 | 146 (26.8) | 54 (28.1) | 92 (26.1) | .512 |
| Mental/physical effect of mind-altering drugs | 32 (5.9) | 17 (8.9) | 15 (4.2) | .028 | 223 (40.9) | 94 (49) | 129 (36.5) | .004 |
| Alcohol, smoking and drug addiction | 39 (7.2) | 25 (13) | 14 (4) | < .001 | 179 (32.8) | 77 (40.1) | 102 (28.9) | .008 |
| Addiction | 17 (3.1) | 9 (4.7) | 8 (2.3) | .141 | 136 (25) | 54 (28.1) | 82 (23.2) | .303 |
| **Growth, Development and Sexual Health** | | | | | | | | |
| Ovulation and menstruation | 58 (10.6) | 20 (10.4) | 38 (10.8) | .980 | 180 (33) | 46 (24) | 134 (38) | .002 |
| Ejaculation | 88 (16.1) | 52 (27.1) | 36 (10.2) | < .001 | 152 (27.9) | 89 (46.4) | 63 (17.8) | < .001 |
| How to use condoms | 100 (18.3) | 60 (31.3) | 40 (11.3) | < .001 | 167 (30.6) | 94 (49) | 73 (20.7) | < .001 |
| Woman's basal body temperature | 13 (2.4) | 12 (6.3) | 1 (0.3) | < .001 | 102 (18.7) | 32 (16.7) | 70 (19.8) | .901 |
| Fertilization and pregnancy | 46 (8.4) | 28 (14.6) | 18 (5.1) | < .001 | 141 (25.9) | 67 (34.9) | 74 (21) | < .001 |
| Gestation period and childbirth | 19 (3.5) | 11 (5.7) | 8 (2.3) | .033 | 127 (23.3) | 50 (26) | 77 (21.8) | .233 |
| Public services related to pregnancy | 7 (1.3) | 4 (2.1) | 3 (0.8) | .198 | 105 (19.3) | 43 (22.4) | 62 (17.6) | .101 |
| Significance and importance of family planning | 10 (1.8) | 7 (3.6) | 3 (0.8) | .018 | 93 (17.1) | 40 (20.8) | 53 (15) | .063 |
| Contraception | 90 (16.5) | 48 (25) | 42 (11.9) | < .001 | 215 (39.4) | 99 (51.6) | 116 (32.9) | < .001 |
| Abortion under Maternal Health Act in Japan | 8 (1.5) | 4 (2.1) | 4 (1.1) | .356 | 132 (24.2) | 48 (25) | 84 (23.8) | .622 |
| Sex and gender | 50 (9.2) | 24 (12.5) | 26 (7.4) | .089 | 168 (30.8) | 72 (37.5) | 96 (27.2) | .051 |
| LGBTQIA+ | 72 (13.2) | 24 (12.5) | 48 (13.6) | .763 | 277 (50.8) | 102 (53.1) | 175 (49.6) | .308 |
| **Personal and Community Health** | | | | | | | | |
| Prevention of sexually transmitted diseases | 37 (6.8) | 26 (13.5) | 11 (3.1) | < .001 | 167 (30.6) | 79 (41.1) | 88 (24.9) | < .001 |
| Pathogens that cause infections | 22 (4) | 14 (7.3) | 8 (2.3) | .004 | 197 (36.1) | 82 (42.7) | 115 (32.6) | .010 |
| Transmission of infection | 25 (4.6) | 19 (9.9) | 6 (1.7) | < .001 | 286 (52.5) | 107 (55.7) | 179 (50.7) | .225 |
| Vaccines | 21 (3.9) | 16 (8.3) | 5 (1.4) | < .001 | 213 (39.1) | 90 (46.9) | 123 (34.8) | .003 |

NOTE: Values in parenthesis represent the percentage of valid N for each category. Results are based on two-sided tests to compare between sex. The p-value threshold for significance was adjusted for multiple comparisons using the Bonferroni correction. The threshold for statistical significance was set at p < .05 (S1 File).

menstruation, how to use condoms, women's basal body temperature, measurement of women's basal body temperature, abortion under the Maternal Health Act in Japan, and sex and gender (Table 4B).

**Table 6. Self-assessment of knowledge score comparison between sexes.**

| Key words | Sex | | | Male | | | Female | | | p |
|---|---|---|---|---|---|---|---|---|---|---|
| *Nutrition and Physical Activity* | Mann-Whitney U | Wilcoxon W | Z | N | Mean Rank | Rank Sum | N | Mean Rank | Rank Sum | |
| Lifestyle-related diseases | 34776 | 96904 | .786 | 190 | 264.5 | 50249 | 352 | 275.3 | 96904 | .432 |
| Food poisoning | 34866 | 97347 | .566 | 192 | 267.9 | 51439 | 353 | 275.8 | 97347 | .571 |
| Allergic reaction | 38638 | 100414 | 2.984 | 191 | 244.7 | 46739 | 351 | 286.1 | 100414 | .003 |
| *Alcohol, Tobacco, and Other Drugs* | | | | | | | | | | |
| Health effect of smoking | 31622 | 94103 | -1.314 | 192 | 284.8 | 54682 | 353 | 266.6 | 94103 | .189 |
| Health hazard of passive smoking | 31031 | 93512 | -1.653 | 192 | 287.9 | 55274 | 353 | 264.9 | 93512 | .098 |
| Smoking effect on pregnancy | 38054 | 99830 | 2.525 | 192 | 249.3 | 47867 | 351 | 284.4 | 99830 | .012 |
| Dependence of smoking | 29867 | 91995 | -2.182 | 191 | 291.6 | 55701 | 352 | 261.4 | 91995 | .029 |
| Health effect of e-cigarettes | 31109 | 93237 | -1.449 | 191 | 285.1 | 54459 | 352 | 264.9 | 93237 | .147 |
| Effect of alcohol consumption | 34666 | 97147 | .555 | 191 | 267.5 | 51093 | 353 | 275.2 | 97147 | .579 |
| Alcohol consumption effect on pregnancy | 35638 | 97414 | 1.339 | 190 | 258.9 | 49197 | 351 | 277.5 | 97414 | .181 |
| Mental/physical effect of mind-altering drugs | 33824 | 95952 | .019 | 192 | 272.3 | 52288 | 352 | 272.6 | 95952 | .985 |
| Alcohol, smoking and drug addiction | 33391 | 95872 | -.287 | 192 | 275.6 | 52913 | 353 | 271.6 | 95872 | .774 |
| Addiction | 32058 | 92436 | -.534 | 190 | 273.8 | 52017 | 347 | 266.4 | 92436 | .593 |
| *Growth, Development and Sexual Health* | | | | | | | | | | |
| Ovulation and menstruation | 56749 | 119230 | 13.155 | 192 | 153.9 | 29556 | 353 | 337.8 | 119230 | < .001 |
| Ejaculation | 16419 | 78900 | -10.037 | 192 | 364.0 | 69886 | 353 | 223.5 | 78900 | < .001 |
| How to use condoms | 16618 | 77344 | -9.668 | 191 | 357.0 | 68186 | 348 | 222.3 | 77344 | < .001 |
| Measurement of woman's basal body temperature | 46965 | 108741 | 8.176 | 188 | 195.7 | 36790 | 351 | 309.8 | 108741 | < .001 |
| Fertilization and pregnancy | 30672 | 92800 | -1.611 | 190 | 286.1 | 54353 | 352 | 263.6 | 92800 | .107 |
| Pregnancy and childbirth | 35016 | 97497 | .860 | 190 | 264.2 | 50199 | 353 | 276.2 | 97497 | .390 |
| Public services related to pregnancy | 36979 | 98755 | 2.008 | 191 | 253.4 | 48398 | 351 | 281.4 | 98755 | .045 |
| Significance and importance of family planning | 35190 | 95568 | 1.306 | 190 | 257.3 | 48886 | 347 | 275.4 | 95568 | .192 |
| Contraception | 27879 | 89655 | -3.189 | 190 | 299.8 | 56957 | 351 | 255.4 | 89655 | .001 |
| Abortion under Maternal Health Act in Japan | 36010 | 97085 | 1.669 | 190 | 255.0 | 48446 | 349 | 278.2 | 97085 | .095 |
| Sex and gender | 29588 | 90663 | -2.179 | 191 | 290.1 | 55408 | 349 | 259.8 | 90663 | .029 |
| LGBTQIA+ | 32848 | 93923 | -.078 | 189 | 270.2 | 51068 | 349 | 269.1 | 93923 | .938 |
| *Personal and Community Health* | | | | | | | | | | |
| Prevention of sexually transmitted diseases | 27177 | 88953 | -3.597 | 190 | 303.5 | 57658 | 351 | 253.4 | 88953 | < .001 |
| Pathogens that cause infections | 27178 | 88603 | -3.467 | 189 | 301.2 | 56927 | 350 | 253.2 | 88603 | < .001 |
| Transmission of infection | 30853 | 92629 | -1.362 | 189 | 282.8 | 53441 | 351 | 263.9 | 92629 | .173 |
| Vaccines | 33252 | 93630 | .376 | 188 | 264.6 | 49750 | 347 | 269.8 | 93630 | .707 |

NOTE: Mann–Whitney U test was used for comparison. The p-value threshold for significance was adjusted for multiple comparisons using the Bonferroni correction. The threshold for statistical significance was set at p < .05 (S1 File).

## Comparison of sources of information by sex

Significantly more male students selected their teacher(s) as sources of information about LGBTQA+. However, significantly more female students selected their parent(s) as the source of information on the six topics. Furthermore, significantly more male students selected their friend(s) as a source of information than female students did for 20 out of the 29 topics.

Additionally, significantly more male students reported the Internet as a source of information than female students for 16 out of 29 topics, while more female students reported the Internet as a source on one topic (Table 5).

## Comparison of self-assessment by sex

The results of the Mann-Whitney U test demonstrated that male students reported significantly higher self-assessment scores on seven out of 29 topics, while female students reported significantly higher scores on five out of 29 topics (Table 6).

## Discussion

This study aimed to assess the knowledge, sources of information, and self-assessment of essential public health topics among university students before beginning their healthcare education, as well as to evaluate differences in the choice of colleges that indicate their desired career path. To the best of our knowledge, there has been no prior investigation into the public health knowledge of healthcare university students before beginning their college education.

Although students reported being familiar of most of the health topics surveyed, there appeared to be a lack of knowledge in certain areas and reliance on skewed information sources, as detailed below.

### Nutrition and physical activity

Ninety percent or more of the students affirmed their awareness of all three questions related to food poisoning and allergies, and reported their parent(s) and the Internet as sources of information. This finding suggests that education on public health knowledge related to daily life is likely to be provided at home. Although many students reported the internet as their source of information, it is important to note that the information available on the internet is not necessarily accurate. This could be particularly critical in situations related to food poisoning or allergies, where inaccurate information could have serious consequences.

### Alcohol, tobacco, and other drugs

Over 75% of students indicated acquiring information concerning alcohol, tobacco, and other substances from both inside and outside the classroom, with the exception of the health impacts of e-cigarettes. The information sources included teachers, parents, and the Internet. However, many students were unaware of the health impact of e-cigarettes and addiction as a disease. They stated that they had never learned about these two topics in school. Although e-cigarettes are relatively new, their negative health effects have been reported in Japan and worldwide [30, 31]. The regulations for conventional cigarettes do not apply to e-cigarettes [32], leading to many advertisements targeting younger generations that present e-cigarettes as a less harmful alternative. We believe that education on this topic is essential in schools.

### Growth, development, and sexual health

Nine out of the 12 topics had 10% or more students who reported having no knowledge, and 10% or more students reported that they had not learned about eight of these topics in class. In particular, fewer female students reported knowing how to use condoms, and fewer male students reported knowing about women's basal temperatures. Furthermore, male students reported the Internet as a source of information on how to use condoms, and female students reported parents as a source of information on the basal temperature of women. In addition, many students reported not having learned about the public services available during

pregnancy, the importance of family planning, biological and social concepts of sex, gender identity, and what LGBTQIA+ stands for in school. They reported friends as the information source for these topics. Despite LGBTQIA+ individuals having specific healthcare requirements, curricula do not address these requirements in depth [33]. It is important for future healthcare workers to receive adequate education regarding sexual health and related public services, as well as LGBTQIA+-related health information, to provide appropriate healthcare to diverse societies.

## Personal and community health

Interestingly, many students reported knowing about infection pathways, vaccination, and essential information during the COVID-19 pandemic, with the Internet having a high percentage as a source of information. It is speculated that this awareness may be due to the COVID-19 pandemic. This finding is consistent with previous reports on knowledge and information sources regarding COVID-19 from around April 2020 in nursing [34], medicine [35], medical and health science [36], and undergraduate and postgraduate students [37]. These previous reports confirmed that students had adequate knowledge of COVID-19, and for the majority of students, the source of information was social media. Furthermore, many students reported that their sources of information about vaccination were their parents, compared with other topics. It can be speculated that vaccination, especially for minors, is a parental decision, even in the case of the COVID-19 vaccination. There have been reports that many people with vaccine hesitancy obtain information from the Internet and celebrities and tend to be suspicious of healthcare providers [38–41]. Therefore, education and communication are crucial [42, 43]. Environmental education studies on university freshmen have reported that the most important source of environmental education is the family, which provides long-term information and influences students' knowledge and attitudes [44]. Thus, although accurate knowledge must be imparted to children in school, parents should also have access to accurate sources of information.

## Overview

This study revealed skewed information sources, with more male students relying on the Internet and friends as information sources and a disparity between sexes in knowledge and self-assessment, especially regarding sexual health. Our results are consistent with a previous study on the source of sexual knowledge, which reported that female students' ideal source of knowledge was school as a primary source, while male students preferred the Internet, friends, and parents in that order [13]. However, the disparity in class involvement for knowledge between the sexes on some sexual health topics in our study is notable. Although more female students reported that they did not learn about "how to use condoms" or "sex and gender" in class, more male students reported that they did not learn about topics related to women's sexual health, such as "women's basal body temperature" or "gestation period and childbirth". We could speculate about several possibilities to explain this disparity. One possibility is actual educational differences based on sex, while the other is perceived differences. Historically, sexual education in Japan was conducted separately based on sex until recently [10, 45, 46]. Due to the long history of sexual education or lack thereof in Japan, students may consider topics not directly connected to their sex as irrelevant to them or not theirs to learn, thinking it is not their concern. With Japan ranking 120[th] out of 156 countries in 2021 in the Global Gender Gap Index, it is crucial to educate not only students but also parents and educators that sexual health education is equally essential regardless of sex. Teaching sex education as health education, rather than merely teaching physiological differences and reproductive functions, is

crucial to raising awareness of infection prevention, especially sexually transmitted infections, as well as enriching students' lives to build a healthy society [47]. However, this survey also demonstrated that practical and essential knowledge of real life was somewhat limited, despite students' knowledge of physiological differences.

This lack of knowledge is likely contributing to the increasing rates of sexually transmitted infections among young people within Japan and worldwide [3, 5]. With the rise of HIV/AIDS in the 1980s [48, 49], the importance of health education, including sex education, came to the forefront. However, political and social background differences have resulted in varying approaches to sexual health education worldwide, with some countries promoting abstinence-only education, lacking sexual health education altogether, or even discouraging its provision.

During the COVID-19 pandemic, educational lectures and practical training were moved to online formats, including healthcare education [50, 51]. While some countries and regions have accelerated medical students' graduation and assigned them to work [52], the role of medical students during the pandemic remains controversial [53]. It is vital for the public, especially healthcare students, to have accurate public health information in this day and age. Thus, understanding these students' knowledge gaps provides an opportunity to supplement and reinforce their understanding of critical health topics.

## Limitations

There are some limitations to this study. First, while we surveyed seven college programs, we only conducted the study at one Japanese university. Thus, we cannot ignore the possibility that the results would be different if the study were conducted at a different university or country. Second, to the best of our knowledge, we were unable to find a validated public health knowledge questionnaire suitable for this study, and hence, we selected topics from the curriculum guidelines and current official textbooks ourselves. To provide anonymity, obtain honest answers, and give students the opportunity to gain insight into their own information gaps, we did not test the students; instead, we had them self-report their knowledge or lack thereof. Thus, there is a possibility of inaccurate reporting of their knowledge. Thirdly, we cannot rule out the possibility that students who answer surveys are not representative of the entire population. Finally, this study was conducted during the COVID-19 pandemic. Although the study provides a glimpse into students' knowledge and attitudes in the age of the pandemic, the results may not be generalizable. Therefore, we recommend conducting further research on public health knowledge, both within and beyond Japan, to better understand the current state of health education and identify areas for improvement.

## Implications

In Japan, health and sexual health education are not considered as enrollment requirements in any college, including health-related colleges, considering that they are not included in entrance exams. Therefore, students seldom receive health education, let alone enough to proceed with higher healthcare education. Additionally, Japanese medical students' choice to attend medical school tends to be extrinsically rather than intrinsically motivated. Thus, they choose medicine because their grades are suitable for medical school admission. Although differences in knowledge on individual topics exist, no significant differences in total knowledge or total self-assessment scores were observed between colleges. Although further research is needed, we can speculate that medical students are not necessarily focusing on learning each topic related to public health towing to the reasons above, and they are not as interested in public health topics as is expected. The health topics in this study are particularly crucial topics derived from preschool curricula and textbooks through the end of high school in Japan and

the United States. Hence, they are not expected to be taught in universities or healthcare-related colleges. In other words, the lack of knowledge at this point may not be supplemented before students become healthcare workers. To address this issue, we believe that supplemental education reinforcing public health knowledge at the beginning of university education is beneficial. However, conventional classroom education may not be effective as students may not consider all topics to be of concern. Instead, we suggest using student-centered learning methods, such as problem-based learning [54], project-based learning [55], and flipped classrooms [56], as additional supplemental basic public health knowledge education after enrollment. We believe that this approach will lay the foundation for future healthcare workers' knowledge and, subsequently, enhance the essential knowledge and awareness required for a healthy society.

## Conclusions

In this cross-sectional study, we identified a significant knowledge gap and an overreliance on misinformation sources concerning public health, particularly sexual health, which varied by gender. This finding emphasizes the need for early public health education and awareness-raising efforts in universities, emphasizing the equal importance of public health knowledge regardless of gender. Japanese education systems primarily focus on preparing students for university entrance exams, which has led to the neglect of health education as it is not a mandatory part of the exam or university admission requirements. Given the widespread misinformation present in today's world, a policy for improving health education is essential not only to address the lack of accurate knowledge and prevent healthcare workers from acquiring erroneous information but also to raise awareness and decrease the gender gap in sexual health. In conclusion, introducing healthcare-related knowledge to university students, particularly as part of freshmen or healthcare-related schools' curricula, would be beneficial for building a healthy society. This approach would run in parallel with professional training and contribute to creating awareness about public health issues, ultimately leading to a better-informed population and a healthier society.

## Supporting information

**S1 Table. List of specific knowledge questions regarding public health topics and answer choices (multiple answers).**
(DOCX)

**S2 Table. Inventory of questionnaire items, along with individual question response rates and corresponding keywords in tables.**
(DOCX)

**S3 Table. Comparison of students' previous knowledge between sexes.**
(DOCX)

**S1 File. Questionnaire raw data.**
(SAV)

## Acknowledgments

The authors express their gratitude to Editage™ for their assistance in editing the manuscript. Additionally, we extend our thanks to the students who participated in this study.

## Author Contributions

**Conceptualization:** Miwa Sekine.

**Data curation:** Miwa Sekine, Yuko Nakanishi, Shinobu Sakurai, Tomomi Iwashimizu, Yasuaki Sakano, Tetsuya Takahashi.

**Formal analysis:** Miwa Sekine, Shuko Nojiri, Yuji Nishizaki.

**Funding acquisition:** Miwa Sekine.

**Investigation:** Miwa Sekine, David Aune, Shuko Nojiri, Makino Watanabe, Yuko Nakanishi, Shinobu Sakurai, Tomomi Iwashimizu, Yasuaki Sakano, Tetsuya Takahashi, Yuji Nishizaki.

**Methodology:** Miwa Sekine, Shinobu Sakurai, Yuji Nishizaki.

**Project administration:** Miwa Sekine.

**Resources:** Miwa Sekine.

**Software:** Yuji Nishizaki.

**Supervision:** Miwa Sekine.

**Validation:** Miwa Sekine, David Aune.

**Visualization:** Miwa Sekine, David Aune.

**Writing – original draft:** Miwa Sekine, David Aune, Yuji Nishizaki.

**Writing – review & editing:** Miwa Sekine, David Aune, Shuko Nojiri, Makino Watanabe, Yuko Nakanishi, Shinobu Sakurai, Tomomi Iwashimizu, Yasuaki Sakano, Tetsuya Takahashi, Yuji Nishizaki.

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
