## [Decision Letter · Decision Letter 0]

13 Jul 2023

PONE-D-23-16475Cross-Sectional Study on Public Health Knowledge among First-Year University Students in Japan: Implications for Educators and Educational InstitutionsPLOS ONE

Dear Dr. Sekine,

Thank you for submitting your manuscript to PLOS ONE. After careful consideration, we feel that it has merit but does not fully meet PLOS ONE’s publication criteria as it currently stands. Therefore, we invite you to submit a revised version of the manuscript that addresses the points raised during the review process. Please see comments below and revise your manuscript accordingly.

We look forward to receiving your revised manuscript.

Kind regards,

Amos Buh, BSc., MPH

Academic Editor

PLOS ONE

Additional Editor Comments:

Please have this manuscript reviewed by a native English speaker for correction of grammatical and typographical errors. Also, revise the manuscript following the reviewer comments.

Reviewers' comments:

Reviewer's Responses to Questions

**Comments to the Author**

1. Is the manuscript technically sound, and do the data support the conclusions?

Reviewer #1: Partly

Reviewer #2: Partly

Reviewer #3: Yes

2. Has the statistical analysis been performed appropriately and rigorously? 

Reviewer #1: Yes

Reviewer #2: Yes

Reviewer #3: Yes

3. Have the authors made all data underlying the findings in their manuscript fully available?

Reviewer #1: Yes

Reviewer #2: Yes

Reviewer #3: Yes

4. Is the manuscript presented in an intelligible fashion and written in standard English?

Reviewer #1: Yes

Reviewer #2: No

Reviewer #3: Yes

5. Review Comments to the Author

Reviewer #1: Although the paper has merit, there is little value of the findings of the study for the wider scientific community. Perhaps a local journal will be interested in the findings of the study. Moreover there is extensive published literature in this area.

Reviewer #2: - The abstract summarises the main findings. Mention the impact of COVID-19 pandemic on misinformation and knowledge disparities in the introduction part as well.

- Methods: Mention the sample size calculation and power analysis used. Also, the methods section needs to be expanded and should include how the participants were recruited into the study and how authors choose to select the sample.

- Results: the table font and size should be consistent and are hard to read. Consider splitting the tables, in particular Table 4.

- Manuscript must to be revised for typos/grammar by a native English speaker.

Reviewer #3: Title: Cross-Sectional Study on Public Health Knowledge among First-Year University

Students in Japan: Implications for Educators and Educational Institutions.

The title is in line with study objectives and the results

Abstract: Not structured

• Line 56 - Response rate: How well does response to one question a good measure of response rate? If this is used as response rate it is desirable to also mention the proportion that responded to the questions and were used for data analysis.

• Line 57 - The results revealed that health knowledge, …… Is it health knowledge or public health knowledge. They are not the same. It is therefore important to be consistent.

Introduction: Detailed information on statement of problem as well as rational for the study clearly presented.

• Line 68 - 1st sentence: Avoid 1 - sentence paragraph

Methods: Fairly well described.

• Line 182 …..total knowledge and self-assessment scores - What is the maximum total score from the survey tool. It is desirable to categorise total score for respondents into poor, fair and good.

•

Result: Well written in details with relevant tables and figures

• Line 200 …… 549 responded - Include the percentage

• Line 208 & 212 - change 'in' to 'among'

• Line 252: … self-assessed knowledge was 7.0 - Out of a maximum score of what?

• Line 302: The use of the word numerous does not in any way give readers a clear idea of the magnitude or level of knowledge. It is more than three-quarter or more than half.

Discussion: The study findings are well discussed, with study limitations provided.

Conclusion: Clearly written with appropriate recommendation.

6. PLOS authors have the option to publish the peer review history of their article (what does this mean?). If published, this will include your full peer review and any attached files.

Reviewer #1: No

Reviewer #2: No

Reviewer #3: **Yes**

---

## [Author Response · Author response to Decision Letter 0]

17 Aug 2023

Manuscript PONE-D-21-35513

Response to Reviewers

Dear Dr. Buh

We extend our sincere gratitude to you for affording us the privilege to submit a revised version of the manuscript titled “Cross-Sectional Study on Public Health Knowledge among First-Year University Students in Japan: Implications for Educators and Educational Institutions” for consideration in PLOS ONE. We appreciate the time and effort you and the reviewers dedicated to providing feedback on our manuscript. Your invaluable insights and constructive comments have significantly enriched the quality of our paper. We have diligently incorporated most of the recommendations proposed by the reviewers, and these changes are highlighted within the manuscript. For a comprehensive overview of our responses to the reviewers’ comments and concerns, please find the detailed point-by-point response highlighted in blue below. Kindly note that all page references refer to the revised manuscript file with tracked changes.

Additional Editor Comments:

Please have this manuscript reviewed by a native English speaker for correction of grammatical and typographical errors. Also, revise the manuscript following the reviewer comments.

Author response: We express our gratitude for this recommendation. The manuscript has been carefully reviewed by an experienced editor, who is proficient in the English language and has specialized expertise in editing papers written by non-native English-speaking scientists.

Reviewers' Comments to the Authors:

Reviewer #1: 

Although the paper has merit, there is little value of the findings of the study for the wider scientific community. Perhaps a local journal will be interested in the findings of the study. Moreover there is extensive published literature in this area.

Author response: We express our gratitude to Reviewer#1 for their insightful comments. While we acknowledge and concur with the reviewer’s feedback regarding the significance of this study for local journals, we believe that the contributions of this study extends beyond regional boundaries. This study holds the potential to make a meaningful impact on the broader academic community, as it delves into the public health knowledge and awareness of first-year university students. The insights garnered from our study can prove invaluable to educators worldwide. Although our findings perhaps emanate from a specific local context, they are designed to offer actionable solutions for educators, not only within this context but also in a more global perspective. These insights can be particularly pertinent during future crises, fostering adaptable approaches rooted in the observations and conclusions drawn from our research. 

 

Reviewer #2: 

Author response: We extend our sincere appreciation to Reviewer #2 for their valuable and insightful suggestions that have helped us to improve our paper considerably. As indicated in the subsequent responses, we have incorporated each of these comments and suggestions into the revised version of our paper (yellow highlighted).

1. The abstract summarizes the main findings. Mention the impact of COVID-19 pandemic on misinformation and knowledge disparities in the introduction part as well.

Author response: We appreciate your suggestion. We have incorporated the impact of COVID-19 pandemic on misinformation and knowledge disparities into the introduction section.

The revised text reads as follows on [page 5, line 111]:

“The disparities in public health knowledge have been accentuated and brought to the forefront amidst the backdrop of the COVID-19 pandemic. This global crisis has magnified the existing gaps and biases, leading to the proliferation of misinformation and exacerbating knowledge disparities. A prominent consequence of this phenomenon has been the emergence of vaccine hesitancy, despite the development and widespread distribution of COVID-19 vaccines.”

2. Methods: Mention the sample size calculation and power analysis used. Also, the methods section needs to be expanded and should include how the participants were recruited into the study and how authors choose to select the sample.

Author response: We appreciate your suggestion. Because this was an exploratory study encompassing the distribution of questionnaires to all first-year students at Juntendo University, we did not determine the sample size or conduct power analysis. We have revised and clarified the methodological aspects relating to sample size calculation, power analysis and participant recruitment in the Method section. 

The revised text reads as follows on [page 7, line 144]:

“We distributed web-based questionnaires to the entire cohort of first-year students (n = 1,562) across seven programs at Juntendo University. We utilized the university’s universal message system to distribute the questionnaires among the different program divisions: Medicine (n = 136), Healthcare and Nursing (n = 204), Health Science and Nursing (n = 127), Health Science/Physical Therapy (n = 121), Health Science/Radiological Technology (n = 121) Health and Sports Science (n = 608), and International Liberal Arts (n = 245). This survey was conducted between April and May 2021. All participants provided informed consent and had the option to decline participation. Prior to their involvement, participants received a comprehensive overview of the study, including detailed information about data management procedures. They were informed that their data would be anonymized and participation was voluntary.” 

The revised text reads as follows on [page 10, line 207]:

“Given the available resources and constraints in data collection, we gathered data from 549 first-year students, representing a subset of the total population of 1,562 first-year students within the university. As such, a formal sample size calculation or power analysis was not conducted because of the exploratory nature of this study and the comprehensive distribution of the questionnaire to the total first-year student population at Juntendo University.”

3. - Results: the table font and size should be consistent and are hard to read. Consider splitting the tables, in particular Table 4.

Author response: We express our gratitude for this suggestion. We have addressed the font size discrepancy for consistency and implemented a split in Table 4, dividing it into Table 4(A) and Table 4(B) to enhance readability. 

The revised tables are on [pages 18–20]:

4. - Manuscript must to be revised for typos/grammar by a native English speaker.

Author response: We appreciate your suggestion. We have had the manuscript language professionally reviewed.

 

Reviewer #3: 

Author response: We extend our sincere gratitude to Reviewer #3 for their insightful comments and valuable suggestions that have helped us to considerably enhance the quality of our paper. As indicated in the subsequent responses, we have diligently incorporated each of these comments and suggestions into the revised version of our paper (blue highlighted).

1. • Line 56 - Response rate: How well does response to one question a good measure of response rate? If this is used as response rate it is desirable to also mention the proportion that responded to the questions and were used for data analysis.

Author response: We appreciate you for this suggestion. We have integrated the response rate for each question into Supplemental table 2, accompanied by corresponding descriptions within the abstract and result sections.

The revised text reads as follows on [page 3, line 55]:

“(participants’ response rate for each question; 59.6%–100%)”

The revised text reads as follows on [page 11, line 229]:

“The participants’ response rate ranged from a minimum of 59.6% to a maximum of 100% (Supplemental Table S2).”

The revised text reads as follows on [Supplemental table 2]:

“Inventory of questionnaire items, along with individual question response rates and corresponding keywords in tables”

2. Line 57 - The results revealed that health knowledge, …… Is it health knowledge or public health knowledge. They are not the same. It is therefore important to be consistent.

Author response: We appreciate you for this suggestion. We have directed our focus toward public health knowledge; hence, we have appropriately rectified the terminologies. Additionally, we have revised the sentence.

The revised text reads as follows on [page 3, line 42]:

“knowledge gaps and biases in public health information”

The text was revised on [page 3, line 58; page 8, line 160; page 24, line 339; page 30, line 475],

3. Line 68 - 1st sentence: Avoid 1 - sentence paragraph.

Author response: We appreciate you for this suggestion. We have revised the first paragraph.

The revised text reads as follows on [page 4, line 66]:

“Public health knowledge is vital for a healthy society. In recent years,...”

4. Line 182 -total knowledge and self-assessment scores - What is the maximum total score from the survey tool. It is desirable to categorise total score for respondents into poor, fair and good.

Author response: We express our gratitude to you for this suggestion. We have incorporated population tables representing three distinct score levels: Poor (below the 25 percentile), Fair (25–75 percentile), and Good (75 percentile). These tables, designated as Table 2-2, have been included in our work. Additionally, we have included corresponding descriptions within the data analysis and results sections to provide comprehensive insights into the findings.

The revised text reads as follows on [page 9, line 191]:

“The maximum score for total knowledge and self-assessment are 29 and 290, respectively. We converted the total knowledge score and self-assessment score into percentage scores, with a maximum attainable score of 100%. To evaluate knowledge level and self-assessment levels across colleges, we categorized scores below the 25th percentile as “Poor,” scores between the 25th and 75th percentile as “Fair,” and scores at or above the 75th percentile as “Good.” We examined the total knowledge and self-assessment scores, as well as the categorization of these scores into three levels (“Poor,” “Fair,” “Good”), using appropriate methods. Specifically, we employed Kruskal–Wallis tests for non-parametric variance analysis, performed one-way analysis of variance for parametric variance assessment, employed Pearson Chi-Square Tests for proportions, and conducted Bonferroni adjusted pairwise comparisons. These analyses aimed to evaluate whether statistically significant differences existed between colleges regarding knowledge and self-assessment levels.”

The revised text reads as follows on [page 11, line 239]:

“Pearson Chi-Square Tests indicated no significant differences among colleges in the three levels of knowledge (p=.122) or self-assessment (p=.635). However, a significant difference was observed among sexes in the three levels of self-assessment in the Health and Sports College (p=.011). Further pairwise comparison revealed significant differences in the “Poor” (p=.010) and “Fair” (p=.006) categories”

The revised tables on [Table 2-1, Table 2-2]

The revised text of table number on [page 11, line 235; page 11, line 239; page 11, line 243]:

5. Line 200-549 responded - Include the percentage.

Author response: We appreciate your suggestion. We have incorporated the percentage.

The revised text reads as follows on [page 11, line 224]:

“549 (35.15%) responded”

6. Line 208 & 212 - change 'in' to 'among'.

Author response: We appreciate your feedback. We have made the necessary revisions as you suggested.

The revised text reads as follows on [page 11, line 234]:

“Knowledge score among”

The revised text reads as follows on [page 11, line 237]:

“total scores among”

7. Line 252: … self-assessed knowledge was 7.0 - Out of a maximum score of what?

Author response: We appreciate you for this suggestion. We have revised the corresponding sentence to clarify the maximum score. 

The revised text reads as follows on [page 21, line 295]:

“the pinnacle of self-assessed knowledge reached 7.0 (95% CI, 7.0–8.0) on a scale with a maximum score of 10.0”

8. Line 302: The use of the word numerous does not in any way give readers a clear idea of the magnitude or level of knowledge. It is more than three-quarter or more than half..

Author response: We appreciate your suggestion. We have addressed terminological corrections and made sentence revisions to enhance clarity.

The revised text reads as follows on [page 24, line 346]:

“Over 75% of students indicated acquiring information concerning alcohol, tobacco, and other substances from both inside and outside the classroom, with the exception of the health impacts of e-cigarettes.”

---

## [Decision Letter · Decision Letter 1]

30 Aug 2023

Cross-Sectional Study on Public Health Knowledge among First-Year University Students in Japan: Implications for Educators and Educational Institutions

PONE-D-23-16475R1

Dear Dr. Sekine,

We’re pleased to inform you that your manuscript has been judged scientifically suitable for publication and will be formally accepted for publication once it meets all outstanding technical requirements.

Kind regards,

Amos Buh, BSc., MPH

Academic Editor

PLOS ONE

Additional Editor Comments (optional):

Reviewers' comments:

Reviewer's Responses to Questions

**Comments to the Author**

1. If the authors have adequately addressed your comments raised in a previous round of review and you feel that this manuscript is now acceptable for publication, you may indicate that here to bypass the “Comments to the Author” section, enter your conflict of interest statement in the “Confidential to Editor” section, and submit your "Accept" recommendation.

Reviewer #2: All comments have been addressed

Reviewer #3: All comments have been addressed

2. Is the manuscript technically sound, and do the data support the conclusions?

Reviewer #2: Yes

Reviewer #3: Yes

3. Has the statistical analysis been performed appropriately and rigorously? 

Reviewer #2: Yes

Reviewer #3: Yes

4. Have the authors made all data underlying the findings in their manuscript fully available?

Reviewer #2: Yes

Reviewer #3: Yes

5. Is the manuscript presented in an intelligible fashion and written in standard English?

Reviewer #2: Yes

Reviewer #3: Yes

6. Review Comments to the Author

Reviewer #2: All the comments has been addressed by the author. Particularly related to the introduction, methodology, and display of results.

Reviewer #3: The authors have adequately corrected all suggestions and corrections mentioned in the first review. The correctios are quite satisfactory.

7. PLOS authors have the option to publish the peer review history of their article (what does this mean?). If published, this will include your full peer review and any attached files.

Reviewer #2: **Yes: **Melodie Al Daccache

Reviewer #3: **Yes: **Prof. Tanimola Makanjuola Akande

---

## [Editor Report · Acceptance letter]

1 Sep 2023

PONE-D-23-16475R1 

Cross-Sectional Study on Public Health Knowledge among First-Year University Students in Japan: Implications for Educators and Educational Institutions 

Dear Dr. Sekine:

I'm pleased to inform you that your manuscript has been deemed suitable for publication in PLOS ONE. Congratulations! Your manuscript is now with our production department. 

Kind regards, 

on behalf of

Mr. Amos Buh 

Academic Editor

PLOS ONE